# Reinforced Active Learning
# for Image Segmentation

**Arantxa Casanova**[*]
École Polytechnique de Montréal
Mila, Quebec Artificial Intelligence Institute
ElementAI

**Pedro O. Pinheiro**
ElementAI

**Negar Rostamzadeh**
ElementAI

**Christopher J. Pal**
École Polytechnique de Montréal
Mila, Quebec Artificial Intelligence Institute
ElementAI

## Abstract

Learning-based approaches for semantic segmentation have two inherent challenges. First, acquiring pixel-wise labels is expensive and time-consuming. Second, realistic segmentation datasets are highly unbalanced: some categories are much more abundant than others, biasing the performance to the most represented ones. In this paper, we are interested in focusing human labelling effort on a small subset of a larger pool of data, minimizing this effort while maximizing performance of a segmentation model on a hold-out set. We present a new active learning strategy for semantic segmentation based on deep reinforcement learning (RL). An agent learns a policy to select a subset of small informative image regions – opposed to entire images – to be labeled, from a pool of unlabeled data. The region selection decision is made based on predictions and uncertainties of the segmentation model being trained. Our method proposes a new modification of the deep Q-network (DQN) formulation for active learning, adapting it to the large-scale nature of semantic segmentation problems. We test the proof of concept in CamVid and provide results in the large-scale dataset Cityscapes. On Cityscapes, our deep RL region-based DQN approach requires roughly 30% less additional labeled data than our most competitive baseline to reach the same performance. Moreover, we find that our method asks for more labels of under-represented categories compared to the baselines, improving their performance and helping to mitigate class imbalance.

## 1 Introduction

Semantic segmentation, the task of labelling an image pixel-by-pixel with the category it belongs to, is critical for a variety of applications such as autonomous driving (Müller et al., 2018; Wang & Pan, 2018), robot manipulation (Schwarz et al., 2018), embodied question answering (Yu et al., 2019) and biomedical image analysis (Ronneberger et al., 2015). Convolutional neural networks (Lecun et al., 1998)-based methods have achieved excellent results on large-scale supervised semantic segmentation, in which we assume pixel-level annotations are available (Farabet et al., 2013; Pinheiro & Collobert, 2014; Long et al., 2015). For such models to work, however, they need a large amount of pixel-level annotations that may require costly human labor (Cordts et al., 2016; Bearman et al., 2016).

Current semantic segmentation datasets have pixel-wise annotations for each image. This standard approach has two important issues: (i) pixel-level labelling is extremely time consuming. For example, annotation and quality control required more than 1.5h per image (on average) on Cityscapes (Cordts et al., 2016), a popular dataset used for benchmarking semantic segmentation methods. (ii) Class imbalance in the data is typically extreme. Certain categories (such as 'building' or 'sky') can appear

---

[*]Work done while interning at ElementAI

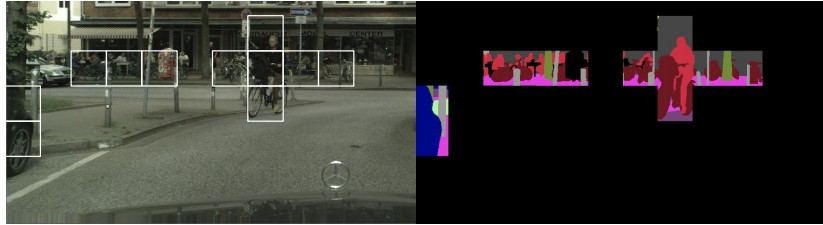

Figure 1: (Left) Input image from Cityscapes dataset (Cordts et al., 2016), with selected regions by our method to be labeled. (Right) Retrieved ground truth annotation for the selected regions. Our method focuses on small objects and under-represented classes, such as bicycles, pedestrians and poles. Best viewed in color.

with two orders of magnitude more frequently than others (*e.g.* 'pedestrian' or 'bicycle'). This can lead to undesired biases and performance properties for learned models.

This is specially relevant when we want to collect annotated data with a human in the loop to create a new dataset or to add more labeled data to an existing one. We can tackle the aforementioned problems by selecting, in an efficient and effective way, which regions of the images should be labeled next. Active learning (AL) is a well-established field that studies precisely this: selecting the most informative samples to label so that a learning algorithm will perform better with less data than a non-selective approach, such as labelling the entire collection of data. Active learning methods can be roughly divided in two groups: (i) methods that combine different manually-designed AL strategies (Roy & McCallum, 2001; Osugi et al., 2005; Gal et al., 2017; Baram et al., 2004; Chu & Lin, 2016; Hsu & Lin, 2015; Ebert et al., 2012; Long & Hua, 2015) and (ii) data-driven AL approaches (Bachman et al., 2017; Fang et al., 2017; Konyushkova et al., 2017; Woodward & Finn, 2016; Ravi & Larochelle, 2018; Konyushkova et al., 2018), that learn which samples are most informative to train a model using information of the model itself. Although label acquisition for semantic segmentation is more costly and time consuming than image classification, there has been considerably less work in active learning for semantic segmentation (Dutt Jain & Grauman, 2016; Mackowiak et al., 2018; Vezhnevets et al., 2012; Konyushkova et al., 2015; Gorriz et al., 2017; Yang et al., 2017), and they focus on hand-crafted strategies.

Current AL techniques that use reinforcement learning (Konyushkova et al., 2018; Fang et al., 2017; Woodward & Finn, 2016; Pang et al., 2018; Padmakumar et al., 2018; Bachman et al., 2017) focus on labelling one sample per step until a budget of labels is met. In semantic segmentation, this would translate into labelling a single region per step. This is highly inefficient, since each step involves updating the segmentation network and computing the rewards. In this work, we propose an end-to-end method to learn an active learning strategy for semantic segmentation with reinforcement learning by directly maximizing the performance metric we care about, *Intersection over Union* (IoU). We aim at learning a policy from the data that finds the most informative regions on a set of unlabeled images and asks for its labels, such that a segmentation network can achieve high-quality performance with a minimum number of labeled pixels. Selecting regions, instead of entire images, allows the algorithm to focus on the most relevant parts of the images, as shown in Figure 1. Although class imbalance in segmentation datasets has been previously addressed in (Badrinarayanan et al., 2017; Chan et al., 2019; Sudre et al., 2017), among others, they try to solve a problem that arises from the data collection process. We show that our proposed method can help mitigate the problem at its source, *i.e.* in the data annotation itself. Because our method maximizes the mean IoU per class, it indirectly learns to ask for more labels of regions with under-represented classes, compared to the baselines. Moreover, we propose and explore a batch-mode active learning approach that uses an adapted DQN to efficiently chose batches of regions for labelling at each step.

To the best of our knowledge, all current approaches for active learning in semantic segmentation rely on hand-crafted active learning heuristics. However, learning a labelling policy from the data could allow the query agent to ask for labeled data as a function of the data characteristics and class imbalances, that may vary between datasets. Our main contributions can be summarized as follows: (i) we learn a RL-based acquisition function for region-based active learning for segmentation, (ii) we formulate our active learning framework with a batch-mode DQN, which labels multiple regions in parallel at each active learning iteration (a more efficient strategy for large-scale datasets that is compatible with standard mini-batch gradient descent), and (iii) we test the proof of concept

in CamVid (Brostow et al., 2008) dataset and provide results in Cityscapes (Cordts et al., 2016) dataset, beating a recent state-of-the-art technique known as BALD (Gal et al., 2017), a widely used entropy-based selection criterion and uniform sampling baselines.

## 2 RELATED WORK

**Active learning**. Traditional active learning techniques focus on estimating the sample informativeness using hand-crafted heuristics derived from sample uncertainty: employing entropy (Shannon, 1948), query-by-committee (Dagan & Engelson, 1995; Shannon, 1948; Freund et al., 1993), maximizing the error reduction (Roy & McCallum, 2001), disagreement between experts (Dagan & Engelson, 1995; Freund et al., 1993) or Bayesian methods that need to estimate the posterior distribution (Houlsby et al., 2011a; Gal et al., 2017). Some approaches combine different techniques to improve AL performance. For instance, relying on exploration-exploitation trade-offs (Osugi et al., 2005), on a bandit formulation (Baram et al., 2004; Chu & Lin, 2016; Hsu & Lin, 2015) and on reinforcement learning (Ebert et al., 2012; Long & Hua, 2015). However, these methods are still limited in the sense that they combine hand-crafted strategies instead of learning new ones. More recent active learning methods rely on an acquisition function that estimates the sample informativeness with a learned metric. Konyushkova et al. (2017) estimate the error reduction of labelling a particular sample, choosing the ones that maximize the error reduction. Wang et al. (2017) introduce a cost-effective approach that also uses confident predictions as pseudo ground truth labels.

**AL with reinforcement learning.** Recently, reinforcement learning has gained attention as a method to learn a labelling policy that directly maximizes the learning algorithm performance. For instance, Liu et al. (2018); Bachman et al. (2017) leverage expert knowledge from oracle policies to learn a labelling policy, and Pang et al. (2018); Padmakumar et al. (2018) rely on policy gradient methods to learn the acquisition function. In a different approach, some methods gather all labeled data in one big step. In Contardo et al. (2017), all samples are chosen in one step with a bi-directional RNN for the task of one-shot learning. In Sener & Savarese (2018), they propose to select a batch of representative samples that maximize the coverage of the entire unlabeled set. However, the bounded core-set loss used tends to perform worse when the number of classes grows.

More similar to our approach, some prior works propose to learn the acquisition function with a Deep Q-Network (DQN) (Mnih et al., 2013) formulation. These works have examined both stream-based active learning (Fang et al., 2017; Woodward & Finn, 2016), where unlabeled samples are provided one by one, and the decision is to label it or not, and pool-based active learning (Konyushkova et al., 2018), where all the unlabeled data is provided beforehand, and the decision is later taken on which samples to choose. The work of Konyushkova et al. (2018) is the closest to ours. Similar to them, our method also leverages the benefits of Q-learning (Watkins & Dayan, 1992) to tackle pool-based AL. Contrary to them, we deal with a much more complex problem: semantic segmentation versus simple classification on UCI repository (Dua & Graff, 2017). The large-scale nature of the problem requires us to use a very different definition of actions, states and rewards. Moreover, we need to adapt the DQN formulation to allow the problem to be computationally feasible.

**AL for semantic segmentation.** Active learning for semantic segmentation has been relatively less explored than other tasks, potentially due to its large-scale nature. For instance, Dutt Jain & Grauman (2016) combine metrics (defined on hand-crafted heuristics) that encourage the diversity and representativeness of labeled samples. Some rely on unsupervised superpixel-based over-segmentation (Vezhnevets et al., 2012; Konyushkova et al., 2015) – and highly depend on the quality of the super-pixel segmentation. Others focus on foreground-background segmentation of biomedical images (Gorriz et al., 2017; Yang et al., 2017), also using hand-crafted heuristics. Settles et al. (2008); Vijayanarasimhan & Grauman (2009); Mackowiak et al. (2018) focus on cost-effective approaches, proposing manually-designed acquisition functions based on the cost of labeling images or regions of images. However, this information is not always given, restricting their applicability.

Mackowiak et al. (2018) focus on cost-effective approaches, where the cost of labeling an image is not considered equal for all images. Similar to our work, they use a region-based approach to cope with the large number of samples on a segmentation dataset. Contrary to us, their labelling strategy is based on manually defined heuristics, limiting the representability of the acquisition function. To the best of our knowledge, our work is the first to apply data-driven RL-based approach to the problem of active learning for semantic segmentation.

## 3  METHOD

We are interested in selecting a small number of regions[1](cropped from images in the original dataset) from a large unlabeled set to maximize the performance of a segmentation network $f$, parameterized by $\theta$. This process is done iteratively until a given budget $B$ of labeled samples is achieved. At each iteration $t$, a query network $\pi$, parameterized by $\phi$, selects $K$ regions to be labeled by an oracle from a large unlabeled set $\mathcal{U}_t$. These samples are added to the labeled set $\mathcal{L}_t$, that is used to train the segmentation network $f$. The performance is measured with a standard semantic segmentation metric, Intersection-over-Union (IoU).

We cast the AL problem within a Markov decision process (MDP) formulation, inspired by other work such as (Padmakumar et al., 2018; Fang et al., 2017; Bachman et al., 2017; Pang et al., 2018; Konyushkova et al., 2018). We model the query network $\pi$ as a reinforcement learning agent, specifically a deep Q-network (Mnih et al., 2013). This data-driven approach allows the model to learn selection strategies based solely on prior AL experience. Our formulation differs from other approaches by the task we address, the definitions of states, actions and rewards, and the reinforcement learning algorithm we use to find the optimal policy.

### 3.1  ACTIVE LEARNING WITH REINFORCEMENT LEARNING FOR SEGMENTATION

In our setting, we use four different data splits. To train $\pi$, we define a subset of labeled data $\mathcal{D}_T$ to *play* the active learning game for several episodes and learn a good acquisition function that maximizes performance with a budget of $B$ regions. The query network is evaluated on a different split $\mathcal{D}_V$. We use a separate subset $\mathcal{D}_R$ to obtain the reward signal by evaluating the segmentation network on it. The set $\mathcal{D}_S$ ($|\mathcal{D}_S| \ll |\mathcal{D}_T|$) is used to construct the state representation.

The MDP is defined with the sequence of transitions $\{(s_t, a_t, r_{t+1}, s_{t+1})\}$. For every *state* $s_t \in \mathcal{S}$ (function of the segmentation network at timestep $t$), the agent can perform *actions* $a_t \in \mathcal{A}$ to choose which samples from $\mathcal{U}_t$ to annotate. The action $a_t = \{a_t^k\}_{k=1}^K$, composed of $K$ sub-actions, is a function of the segmentation network, the labeled and the unlabeled set. Each sub-action asks for a specific region to be labeled. Then, it receives a *reward* $r_{t+1}$ based on the improvement in mean IoU per class after training the segmentation network with the selected samples. Note that states and actions do not depend on the specific architecture of the segmentation network. We are interested in finding a policy to select samples that maximize the segmentation performance. We use deep Q-network (Mnih et al., 2013) and samples from an experience buffer $\mathcal{E}$ to train the query network $\pi$.

Each episode $e$ elapses a total of $T$ steps. We start by setting the segmentation network $f$ to a set of initial weights $\theta_0$ and with no annotated data, *i.e.*, $\mathcal{L}_0 = \emptyset$ and $\mathcal{U}_0 = \mathcal{D}_T$. At each iteration $t$, the following steps are done:

1. The state $s_t$ is computed as function of $f_t$ and $\mathcal{D}_S$.
2. A restricted action space is built with $K$ pools $\mathcal{P}_t^k$ with $N$ regions, sampled uniformly from the unlabeled set $\mathcal{U}_t$. For each region in each pool, we compute its sub-action representation $a_t^{k,n}$.
3. The query agent selects $K$ sub-actions $\{a_t^k\}_{k=1}^K$ with $\epsilon$-greedy policy. Each sub-action $a_t^k$ is defined as selecting one region $x_k$ (out of $N$) to annotate from a pool $\mathcal{P}_t^k$.
4. An oracle labels the regions and the sets are updated: $\mathcal{L}_{t+1} = \mathcal{L}_t \cup \{(x_k, y_k)\}_{k=1}^K$ and $\mathcal{U}_{t+1} = \mathcal{U}_t \setminus \{x_k\}_{k=1}^K$.
5. The segmentation network $f_{t+1}$ is trained one iteration on the recently added regions $\{x_k\}_{k=1}^K$.
6. The agent receives the reward $r_{t+1}$ as the difference of performance between $f_{t+1}$ and $f_t$ on $\mathcal{D}_R$.

Figure 2 depicts this training algorithm. We consider the termination of each episode when the budget $B$ of labeled regions is met, *i.e.*, $|\mathcal{L}_t| = B$. Once the episode is terminated, we restart the weights of the segmentation network $f$ to the initial weights $\theta_0$, set $\mathcal{L}_0 = \emptyset$ and $\mathcal{U}_0 = \mathcal{D}_T$, and restart the episode. We train the query policy $\pi$ by simulating several episodes and updating its weights at each

---

[1]We chose non-overlapping squares as regions (similar to (Mackowiak et al., 2018)). Other choices of regions could also be valid, but we consider region design choice selection to be out of the scope of this work.

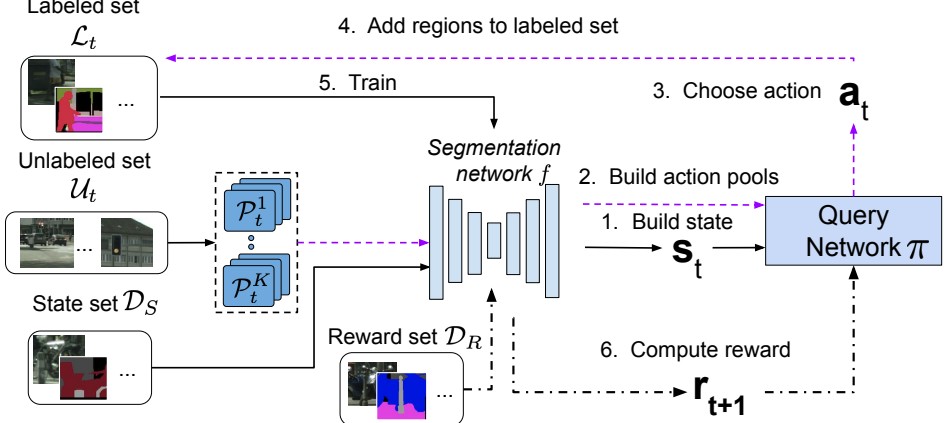

Figure 2: The query network $\pi$ is trained during several episodes $e$ with MDP transitions $\{(s_t, a_t, r_{t+1}, s_{t+1})\}$. 1) The state $s_t$ is computed as a function of segmentation network $f$ and state set $\mathcal{D}_S$. 2) $K$ unlabeled pools $\mathcal{P}_t^k$ are sampled uniformly from the unlabeled set $\mathcal{U}_t$. The representation of their possible sub-actions are computed using $f$, labeled set $\mathcal{L}_t$ and unlabeled set $\mathcal{U}_t$. 3) Query network $\pi$ selects action $a_t$, composed of $K$ sub-actions $a_t^k$. Each of them is chosen from its corresponding pool. 4) Selected regions are labeled and added to $\mathcal{L}_t$ (and removed from $\mathcal{U}_t$). 5) Segmentation network $f$ is trained with those new labeled samples. 6) Reward $r_{t+1}$ is obtained from $\mathcal{D}_R$. This loop continues until a budget $B$ of labeled regions is achieved.

timestep by sampling transitions $\{(s_t, a_t, r_{t+1}, s_{t+1})\}$ from the experience replay buffer $\mathcal{E}$. More details in Section 3.2.

**State representation.**   We would like to use the state of the segmentation network $f$ as the MDP state. Unfortunately, it is not straightforward to embed $f$ into a state representation. Following Konyushkova et al. (2017), we represent the state space $\mathcal{S}$ with the help of a set-aside set $\mathcal{D}_S$. We use a small subset of data from the train set, making sure it contains a significant representation of all classes. We consider it to be a representative set of the dataset, and that any improvement in the segmentation performance on subset $\mathcal{D}_S$ will translate into an improvement over the full dataset[2]. We use the predictions of the segmentation network $f_t$ on $\mathcal{D}_S$ to create a global representation of state $s_t$ (step 1 in Figure 2).

We need a compact representation to avoid intensive memory usage due to the pixel-wise predictions. The samples in $\mathcal{D}_S$ are split in patches, and compact feature vectors are computed for all of them. Then, each region is encoded by the concatenation of two sets of features: one is based on class predictions of $f_t$ and the other on its prediction uncertainty, represented as the Shannon entropy (Shannon, 1948). The first set of features (i) is a (normalized) count of the number of pixels that are predicted to each category. This feature encodes the segmentation prediction on a given patch while dismissing the spatial information, less important for small patches. Moreover, we measure the uncertainty of the predictor with the entropy over the probability of predicted classes. For each region, we compute the entropy of each pixel location to obtain a spatial entropy map. To compress this representation, we apply min, average and max-poolings to the entropy map to obtain downsampled feature maps. The second set of features (ii) is thus obtained by flattening these entropy features and concatenating them.

Finally, the state $s_t$ is represented by an ensemble of the feature representation of each region in $\mathcal{D}_S$. Figure A.1a illustrates how $s_t$ is computed from each region.

**Action representation.**   In our setting, taking an action means asking for the pixel-wise annotation of an unlabeled region. Due to the large-scale nature of semantic segmentation, it would be prohibitively expensive to compute features for each region in the unlabeled set at each step. For this reason, instead, at each step $t$, we approximate the whole unlabeled set by sampling $K$ pools of unlabeled regions $\mathcal{P}_t^k$, each containing $N$ (uniformly) sampled regions. For each region, we compute its sub-action representation $a_t^{k,n}$ (step 2 in Figure 2).

---

[2]In practice, we found that the state set needs to have a similar class distribution as that of the train set.

Each sub-action $a_t^{k,n}$ is a concatenation of four different features: the entropy and class distribution features (as in the state representation), a measure of similarity between the region $x_k$ and the labeled set and another between the region and the unlabeled set. The intuition is that the query network could learn to build a more class-balanced labeled set while still taking representative samples from the unlabeled set. This could help mitigate the hard imbalance of the segmentation datasets and improve overall performance.

For each candidate region, $x$ in a pool $\mathcal{P}_t^k$, we compute the KL divergence between the class distributions of the prediction map of region $x$ (estimated as normalized counts of predicted pixels in each category) and the class distributions of each labeled and unlabeled regions (using the ground-truth annotations and network predictions, respectively). For the labeled set, we compute a KL divergence score between each of the labeled regions' class distribution and the one of region $x$. Summarizing all these KL divergences could be done by taking the maximum or summing them. However, to obtain more informative features, we compute a normalized histogram of KL divergence scores, resulting in a distribution of similarities. As an example, if we were to sum all the scores, having half of the labeled regions with a KL divergence of zero and the other half with a value $c$, would be equivalent to have all labeled regions with a KL divergence of $c/2$. The latter could be more interesting, since it means there are no labeled regions with the same class distribution as $x$. For the unlabeled set we follow the same procedure, resulting in another distribution of KL divergences. Both of them are concatenated and added to the action representation. Figure A.1b illustrates how we represent each possible action in a pool.

Based on early experimentation, learning the state and action representations directly with a CNN does not provide strong enough features for the reinforcement learning framework to converge to a good solution. An ablation study on the state and action components can be found in Appendix E.1.

## 3.2 BATCH MODE DQN

The desired query agent should follow an optimal policy. This policy maps each state to an action that maximizes the expected sum of future rewards. We rely on a DQN (Mnih et al., 2013), parameterized by $\phi$, to find an optimal policy.

We train our DQN with a labeled set $\mathcal{D}_T$ and compute the rewards in a held-out split $\mathcal{D}_R$. As mentioned above, the query agent in our method selects $K$ regions before transitioning to the next state. We assume that each region is selected independently, as in the case where $K$ annotators label one region in parallel. In this case, the action $a_t$ is composed of $K$ independent sub-actions $\{a_t^k\}_{k=1}^K$, each with a restricted action space, avoiding the combinatorial explosion of the action space. To ease computation and avoid selecting repeated regions in the same time-step, we restrict each sub-action $a_t^k$ to select a region $x_k$ in $\mathcal{P}_t^k$ defined as:

$$a_t^k = \underset{a_t^{k,n} \in \mathcal{P}_t^k}{\operatorname{argmax}} \ Q(s_t, a_t^{k,n}; \phi) \ , \tag{1}$$

for each $k \in \{1, ..., K\}$ action take in timestep $t$.

The network is trained by optimizing a loss based on temporal difference (TD) error (Sutton, 1988). The loss is defined as the expectation over decomposed transitions $\mathcal{T}_k = \{(s_t, a_t^k, r_{t+1}^k, s_{t+1})\}$, obtained from the standard transitions $\{(s_t, a_t, r_{t+1}, s_{t+1})\}$, by approximating $r_{t+1}^k \approx r_{t+1}$:

$$\mathbb{E}_{\mathcal{T}_k \sim \mathcal{E}}\left[(y_t^k - Q(s_t, a_t^k; \phi))^2\right] \ , \tag{2}$$

where $\mathcal{E}$ is the experience replay buffer and $y_t^k$ the TD target for each sub-action.

To stabilize the training, we used a target network with weights $\phi'$ and the double DQN (Van Hasselt et al., 2016) formulation. The action selection and evaluation is decoupled; the action is selected with the target network and is evaluated with the query network. The TD target for each sub-action is represented as:

$$y_t^k = r_{t+1} + \gamma Q(s_{t+1}, \underset{a_{t+1}^{k,n} \in \mathcal{P}_{t+1}^k}{\operatorname{argmax}} \ Q(s_{t+1}, a_{t+1}^{k,n}; \phi'); \phi) \ . \tag{3}$$

where $\gamma$ is a discount factor. [3]

---

[3]We also tried to optimize the network with an average of Q-values over all sub-actions as in $y = r_{t+1} + \frac{1}{K}\sum_k Q(s_{t+1}, a_{t+1}^k)$ and $Q(s_t, a_t) = \frac{1}{K}\sum_k Q(s_t, a_t^k)$, but it performed worse.

This formulation is valid under the approximation that the sub-actions are independent of each other, conditioned on the state. We observed that increasing the number of sub-actions $K$ per step eases computation and does not hinder segmentation performance. We provide an ablation study on the effect of $K$ in Appendix E.3.

## 4 EXPERIMENTS

We start this section by describing the datasets that we use to evaluate our method, the experimental setup, and the baselines. We evaluate the algorithm in Camvid as a proof of concept and we show large-scale results on Cityscapes.

### 4.1 EXPERIMENTAL SETUP

Although we can apply active learning in a setting with unlabeled data with a human in the loop that labels selected regions, we test our approach in fully labeled datasets, where it is easier to mask out the labels of a part of the data and reveal them when the active learning algorithm selects them.

**CamVid (Brostow et al., 2008).** This dataset consists of street scene view images, with the resolution of 360×480 and 11 categories. It has 370, 104 and 234 images for train, validation and test set, respectively. We split the train set with uniform sampling in 110 labeled images (from where we get 10 images to represent the state set $\mathcal{D}_S$ and the rest for $\mathcal{D}_T$), and 260 images to build $\mathcal{D}_V$, where we evaluate and compare our acquisition function to the baselines. The state set is chosen to be representative of $\mathcal{D}_T$, by restricting the sampling of $\mathcal{D}_S$ to have a similar class distribution to the one of $\mathcal{D}_T$. Each image is split into 24 regions of dimension 80×90. We use the dataset's validation set for $\mathcal{D}_R$. We report the final segmentation results on the test set. In our experiments, we chose $K = 24$ regions per step. Our model is quite robust to the number of regions selected at each time step (see Appendix E.3).

**Cityscapes (Cordts et al., 2016).** It is also composed of real street scene views, with image resolution of 2048×1024 and 19 semantic categories. The train set with fine-grained segmentation labels has 2975 images and the validation dataset of 500 images. We uniformly sampled 360 labeled images from the train set. Out of these, 10 images represent $\mathcal{D}_S$, 150 build $\mathcal{D}_T$ and 200, $\mathcal{D}_R$, where we get our rewards. The remaining 2615 images of train set are used for $\mathcal{D}_V$, as if they were unlabeled. We report the results in the validation set (test set not available). Each image is split in 128 regions of dimension 128×128. We chose $K = 256$ regions per step.

**Implementation details.** The split $\mathcal{D}_R$ is used to get the rewards for the DQN and also for hyperparameter selection, that are chosen according to the best setup for both baselines and our method. We report the average and standard deviation of the 5 different runs (5 random seeds). As data augmentation, we use random horizontal flips and random crops of $224 \times 224$. For more details, please refer to Appendix B on supplementary material.

**Evaluation.** The query network $\pi$ is trained on $\mathcal{D}_T$ with a small, fixed budget (0.5k regions for Camvid and 4k regions for Cityscapes) to encourage picking regions that will boost the performance in an heavily scarce data regime. The learned acquisition function, as well as the baselines, is evaluated on $\mathcal{D}_V$, where we ask for labels until the budget is met, for different budgets. Note that the baselines do not have any learnable component.

Once the budget is reached, we train the segmentation network $f$ with $\mathcal{L}_T$ until convergence (with early stopping in $\mathcal{D}_R$). For a fair comparison, all methods' segmentation network has been pre-trained (initial $f$ weights $\theta_0$) on GTA dataset (Richter et al., 2016), a synthetic dataset where high amounts of labeled data can be obtained without human effort, and $\mathcal{D}_T$ (where we had labels to train the DQN). We evaluate the final segmentation performance (measured in mean IoU) on the test set of CamVid and on the validation set of Cityscapes.

### 4.2 RESULTS

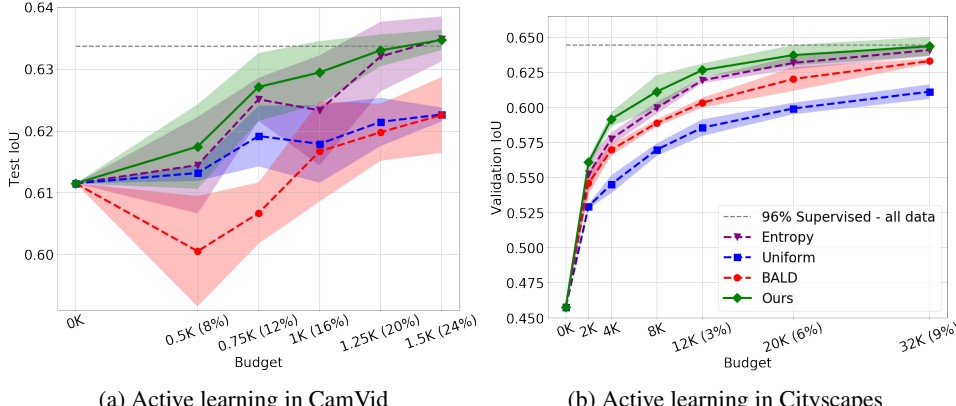

(a) Active learning in CamVid  (b) Active learning in Cityscapes

Figure 4: Performance of several methods with increasing active learning budget, expressed as the number of 128×128 pixel regions labeled and the % of additional labeled data. All methods have been pretrained with GTAV and a small subset of their target datasets. Budget indicates additional number of regions labeled (and the percentage of unlabeled data used). The dashed line represents the 96% of the total performance achieved by the segmentation network with fully-supervised training (having access to all labels). We report the mean and standard deviation of 5 runs.

**Results in CamVid.** We compare our results against three distinct baselines: (i) **U** is the uniform random sampling of the regions to label at each step out of all possible regions in the pool, (ii) **H** is an uncertainty sampling method that selects the regions with maximum cumulative pixel-wise Shannon entropy, (iii) **B** picks regions with maximum cumulative pixel-wise BALD (Houlsby et al., 2011b; Gal et al., 2017) metric. We use 20 iterations of MC-Dropout (Gal & Ghahramani, 2016) (instead of 100, as in (Gal et al., 2017)) for computational reasons. In preliminary experiments, we did not observe any improvement using over 20 iterations. In Camvid, we use a pool size of 10 for our method, **H**, **B** and 50 for **U**. In Cityscapes, we have access to more data so we use pool sizes of 500, 200, 200 and 100 respectively for **U**, **H**, **B** and our method. Pool sizes were selected according to the best validation mean IoU.

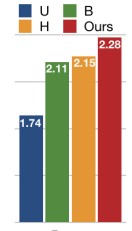

Figure 3: Entropy of class distributions obtained from pixels of selected regions.

Figure 4a shows results on CamVid for different budget sizes. Our method outperforms the baselines for every fixed budget, except for 1.5k regions, where we achieve similar performance as **H**. We argue that the dataset has a small number of images and selecting 1.5k regions already reaches past 98% of maximum performance, where differences between our method and **H** are negligible. Surprisingly, **B** is worse than **U**, specially for small budgets, where training with the newly acquired labels does not provide any additional information. It overfits quickly to the training, getting a worst result that with the initial weights. In general, all results have a high variance due to the low regime of data we are working in. In Appendix E.2 we show the advantages of labeling small regions instead of full images.

**Results in Cityscapes.** Figure 4b shows results on Cityscapes for different budgets. Here, we also observe that our method outperforms the baselines for all budgets points. Labelling 20k regions, corresponding to only 6% of the total pixels (additional to the labeled data in $\mathcal{D}_T$), we obtain a performance of 64.5% mean IoU. This is 96% of the performance of the segmentation network if it had access to all labeled pixels. To reach the same performance, **H** requires an additional 6k labeled regions (around 30% more pixels, equivalent to an extra 45 images). In this larger dataset, **B** performs better than random, showing that for the task of segmentation, **B** might start to show its benefits only for considerably large budgets. Table 1 shows the per-class IoU for the evaluated methods (with a fixed budget). Our method works specially well for under-represented classes, such as Person, Motorcycle or Bicycle, among others. Indeed, our method selects more pixels belonging to under-represented classes than baselines. Note that this is a side effect of directly optimizing for the mean IoU and defining class-aware representations for states and actions. Figure 3 shows the entropy of the distribution of selected pixels of the final labeled set (for a budget of 12k regions) for Cityscapes. The higher the entropy means closer to uniform distribution over classes, and our method

| Method | Road | Side-Walk | Build-ing | Wall | Fence | Pole | Traffic Light | Traffic Sign | Vege-tation | Terrain |
|---|---|---|---|---|---|---|---|---|---|---|
| U | **96.67** | **76.63** | 88.48 | 33.89 | 36.00 | 52.80 | 54.27 | 60.84 | 90.27 | **52.34** |
| H | 95.60 | 72.08 | 88.06 | **35.30** | **44.59** | 52.43 | 53.70 | 61.38 | 90.08 | 51.87 |
| B | 95.25 | 69.37 | **88.75** | 32.28 | 44.36 | **53.81** | **58.84** | **64.79** | 90.27 | 50.51 |
| Ours | 96.19 | 74.24 | 88.46 | 33.56 | 42.28 | 53.28 | 57.18 | 63.61 | 90.20 | 51.84 |

| | Sky | Person | Rider | Car | Truck | Bus | Train | Motor-cycle | Bicycle | mIoU |
|---|---|---|---|---|---|---|---|---|---|---|
| U | 92.57 | 69.66 | 31.82 | 90.13 | 27.04 | 43.41 | 23.30 | 32.98 | 63.64 | 58.78 |
| H | 88.27 | 72.69 | 40.85 | 90.46 | **42.40** | **58.88** | 33.63 | 43.17 | 68.08 | 62.29 |
| B | **93.33** | 71.16 | 39.08 | 88.38 | 34.23 | 43.41 | 30.35 | 37.37 | 66.67 | 60.64 |
| Ours | 91.32 | **73.30** | **45.22** | **90.91** | 42.14 | 58.84 | **35.97** | **45.14** | **69.35** | **63.32** |

Table 1: Per category IoU and mean IoU [%] on Cityscapes validation set, for a budget of 12k regions. For clarity, only the mean of 5 runs is reported. Results with standard deviations in Table C.1.

has the highest entropy. Appendix C shows the distribution from which the entropy is computed and Appendix D presents some qualitative results, showing what each method decides to label for some images.

## 5 CONCLUSION

We propose a data-driven, region-based method for active learning for semantic segmentation, based on reinforcement learning. The goal is to alleviate the costly process of obtaining pixel-wise labels with a human in the loop. We propose a new modification of DQN formulation to learn the acquisition function, adapted to the large-scale nature of semantic segmentation. This provides a computationally efficient solution that uses less labeled data than competitive baselines, while achieving the same performance. Moreover, by directly optimizing for the per-class mean IoU and defining class-aware representations for states and actions, our method asks for more labels of under-represented classes compared to baselines. This improves the performance and helps to mitigate class imbalance. As future work, we highlight the possibility of designing a better region definition, that could help improve the overall results, and adding domain adaptation for the learnt policy, to transfer it between datasets.

## ACKNOWLEDGEMENTS

We thank NSERC and PROMPT. We would also like to thank the team at ElementAI for supporting this research and providing useful feedback.

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

## A  STATE AND ACTION REPRESENTATION DETAILS

In this section, we provide illustrations that show more details on how the state and action are built. Figure A.1a shows how to build the state representation and Figure A.1b how to compute the action representation of a particular region.

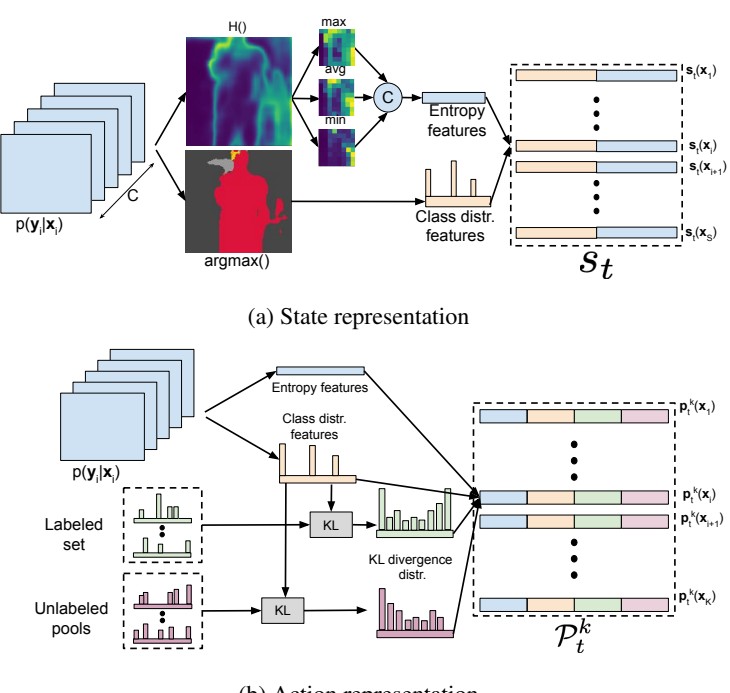

(a) State representation

(b) Action representation

Figure A.1: (a) Each region $x_i$ in $\mathcal{D}_S$ is represented as a concatenation of two features, one based on entropy and the other on class predictions. The final state $s_t$ is the concatenation of the features for all regions. (b) Each region $x_k$ in pool $\mathcal{P}_k$ is represented as a concatenation of four features: entropy-based features, class predictions and two KL divergence distributions, comparing each region $x_k$ with the labeled and unlabeled set.

## B  EXTENDED EXPERIMENTAL SETUP

The segmentation network $f$ is an adaptation of feature pyramid network (Lin et al., 2017) for semantic segmentation (similar to the segmentation branch of (Kirillov et al., 2019)), with a ResNet-50 backbone (He et al., 2016), pretrained on ImageNet (Deng et al., 2009). The network is pretrained on the full train set of a large-scale synthetic dataset, GTAV (Richter et al., 2016), therefore not requiring much human labelling effort. Moreover, this dataset has the advantage of possessing the same categories as real datasets we experiment with.

The query network $\pi$, depicted in Figure B.1, is composed of two paths, one to compute state features and another to compute action features, fusing them at the end. Each of the layers are composed of Batch Normalization, ReLU activation and a fully-connected layer. The state path and action path are composed of 4 and 3 layers, respectively, with a final layer that fuses them together to get the global features; these are gated with a sigmoid, controlled by the KL distance distributions in the action representation. The weights are updated at each step of the active learning loop, by sampling batches of 16 experience tuples from an experience replay buffer, sized 600 and 3200 for Camvid and Cityscapes, respectively.

We train both networks with stochastic gradient descent (SGD) with momentum. We use the same learning rate for both the segmentation and query networks; $10^{-4}$ and $10^{-3}$ for Cityscapes and Camvid respectively. Weight decay is set to $10^{-4}$ for the segmentation network and $10^{-3}$ for the query network. We used a training batch size of 32 for Camvid and 16 for Cityscapes.

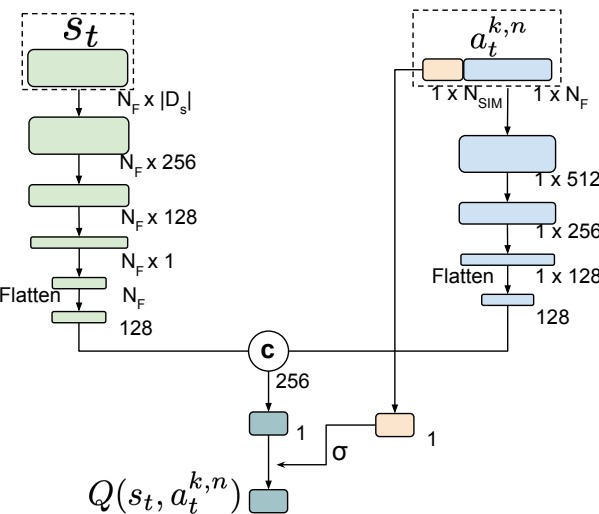

Figure B.1: The DQN takes a state representation $s_t$ and an action representation for a possible action (labeling region $x_k$ in an unlabeled pool $\mathcal{P}_t^k$). $N_F$ are the number of state and action features (class distributions and entropy-based features), and $N_{SIM}$ the number of features for the KL divergence distributions. Features are computed for both representations separately with layers composed of Batch Normalization, ReLU activation and fully connected layers. Both feature vectors are flattened and concatenated, to apply a final linear layer that obtains a score as a single scalar. The Q-values are computed as the gated score, where the gate is controlled by a feature representation from the KL distance distributions of the action representation.

## C    CLASS FREQUENCIES AND PERFORMANCE PER CLASS

We show in Figure C.1 a more detailed plot for the class frequencies of regions that each of the methods chooses for labeling. As the entropy of the class distributions in Figure 3 show, our method picks more regions containing under-represented classes. Specially, it asks labels for more Person, Rider, Train, Motorcycle and Bicycle pixels.

We observe that our **B** baseline picks more than 50% of pixels for only 3 classes that are over-represented or have a medium representation: Building, Vegetation and Sky. This could explain why the performance is worse than the **H** baseline.

Moreover, in table C.1, we extend Table 1 by adding the standard deviation for each result.

## D    QUALITATIVE RESULTS

We compare our method qualitatively with the baselines in Figure D.1. Baseline **U** asks for random patches. Our method tends to pick more regions with the under-represented classes and small objects. For instance, in the first image in the left, our method asks for several regions of a Train, that almost has no samples in the training data. In the second image, it focuses on Person, Bicycle and Poles. In the third image, it asks for labels of the traffic lights and a pedestrian on a bicycle. Baselines **B** and **H** select some of those relevant regions, but miss a lot of them.

## E    ABLATION STUDIES

In this section, we provide an ablation study on the state and action representation, the effect of labeling small regions versus full images, and the comparison of taking different regions per step.

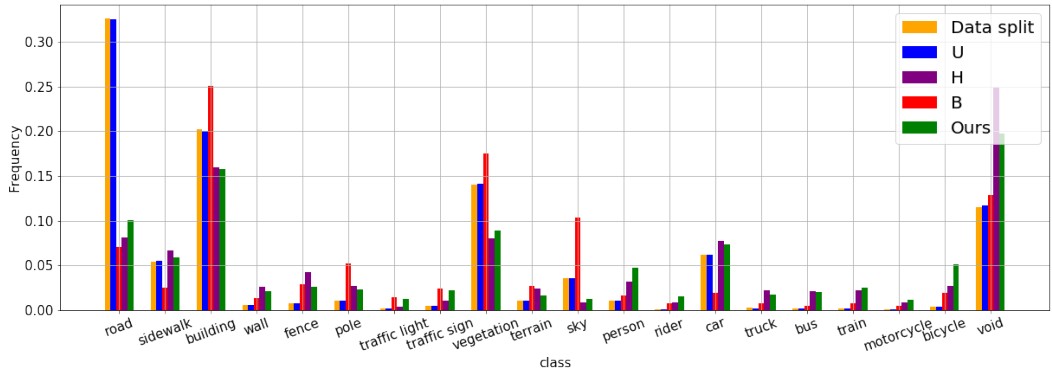

Figure C.1: Class frequencies [%] in Cityscapes for the selected regions to label after the active learning acquisition for different methods. "Data split" frequencies refer to the proportion of classes in the unlabeled data split, where we reveal the masks for the purpose of showing the underlying class frequencies. In this split is where all methods perform active learning, in the setting where we mask out the labels ($\mathcal{D}_v$). Budget used: 12k regions. For ease of visualization, we only plot the mean of 5 runs. Void label represents all pixels for which we do not assign any labels.

| Method | Road | Sidewalk | Building | Wall | Fence |
|---|---|---|---|---|---|
| U | **96.67** $\pm$ 0.09 | **76.63** $\pm$ 0.51 | 88.48 $\pm$ 0.12 | 33.89 $\pm$ 1.11 | 36.00 $\pm$ 2.12 |
| H (Shannon, 1948) | 95.60 $\pm$ 0.33 | 72.08 $\pm$ 1.28 | 88.06 $\pm$ 0.42 | **35.30** $\pm$ 1.73 | **44.59** $\pm$ 1.62 |
| B (Gal et al., 2017) | 95.25 $\pm$ 0.28 | 69.37 $\pm$ 0.94 | **88.75** $\pm$ 0.18 | 32.28 $\pm$ 0.88 | 44.36 $\pm$ 1.12 |
| Ours | 96.19 $\pm$ 0.23 | 74.24 $\pm$ 1.50 | 88.46 $\pm$ 0.23 | 33.56 $\pm$ 2.30 | 42.28 $\pm$ 1.40 |
| | **Pole** | **Traffic Light** | **Traffic Sign** | **Vegetation** | **Terrain** |
| U | 52.80 $\pm$ 0.41 | 54.27 $\pm$ 1.34 | 60.84 $\pm$ 0.99 | 90.27 $\pm$ 0.14 | **52.34** $\pm$ 1.38 |
| H (Shannon, 1948) | 52.43 $\pm$ 0.31 | 53.70 $\pm$ 1.48 | 61.38 $\pm$ 0.81 | 90.08 $\pm$ 0.16 | 51.87 $\pm$ 0.79 |
| B (Gal et al., 2017) | **53.81** $\pm$ 0.30 | **58.84** $\pm$ 0.50 | **64.79** $\pm$ 0.34 | 90.27 $\pm$ 0.15 | 50.51 $\pm$ 0.94 |
| Ours | 53.28 $\pm$ 0.51 | 57.18 $\pm$ 1.92 | 63.61 $\pm$ 1.47 | 90.20 $\pm$ 0.26 | 51.84 $\pm$ 1.62 |
| | **Sky** | **Person** | **Rider** | **Car** | **Truck** |
| U | 92.57 $\pm$ 0.30 | 69.66 $\pm$ 0.62 | 31.82 $\pm$ 2.66 | 90.13 $\pm$ 0.01 | 27.04 $\pm$ 2.16 |
| H (Shannon, 1948) | 88.27 $\pm$ 3.26 | 72.69 $\pm$ 0.53 | 40.85 $\pm$ 1.85 | 90.46 $\pm$ 0.38 | **42.40** $\pm$ 1.96 |
| B (Gal et al., 2017) | **93.33** $\pm$ 0.24 | 71.16 $\pm$ 0.47 | 39.08 $\pm$ 1.26 | 88.38 $\pm$ 0.29 | 34.23 $\pm$ 1.24 |
| Ours | 91.32 $\pm$ 1.06 | **73.30** $\pm$ 0.43 | **45.22** $\pm$ 2.75 | **90.91** $\pm$ 0.23 | 42.14 $\pm$ 1.41 |
| | **Bus** | **Train** | **Motorcycle** | **Bicycle** | **mIoU** |
| U | 43.41 $\pm$ 2.80 | 23.30 $\pm$ 2.52 | 32.98 $\pm$ 3.81 | 63.64 $\pm$ 0.33 | 58.78 $\pm$ 0.29 |
| H  (Shannon, 1948) | **58.88** $\pm$ 2.97 | 33.63 $\pm$ 4.76 | 43.17 $\pm$1.37 | 68.08 $\pm$ 0.38 | 62.29 $\pm$ 0.55 |
| B (Gal et al., 2017) | 43.41 $\pm$4.34 | 30.35 $\pm$ 3.18 | 37.37 $\pm$ 0.79 | 66.67 $\pm$ 0.67 | 60.64 $\pm$ 0.49 |
| Ours | 58.84 $\pm$ 4.15 | **35.97** $\pm$ 3.50 | **45.14** $\pm$ 2.34 | **69.35** $\pm$ 0.90 | **63.32** $\pm$ 0.93 |

Table C.1: Per category IoU and mean IoU [%], on Cityscapes validation set, for a budget of 12k regions. Both the mean and standard deviation of 5 runs is reported.

## E.1 STATE AND ACTION REPRESENTATION

Here, we analyze the incremental effect of our design choices for the state and action representation on Cityscapes. We use 3 pooling operations – min, average, max – to compress the entropy map of the region and use it in the state and action representation. Also, KL divergences are added to the latter. As seen in Table E.1, using only the max-pooled entropy map (**Ours - 1H**), the performance is slightly worse than **H**. When we combine the information of the 3 pooled entropy maps (**Ours - 3H**), we outperform **H** baseline. Moreover, when adding the two distribution of KL distances to our action representation (**Ours - 3H + KL**): between possible regions to label and the labeled set and between the region and the unlabeled set, we further increase the performance, getting our best state and action representations.

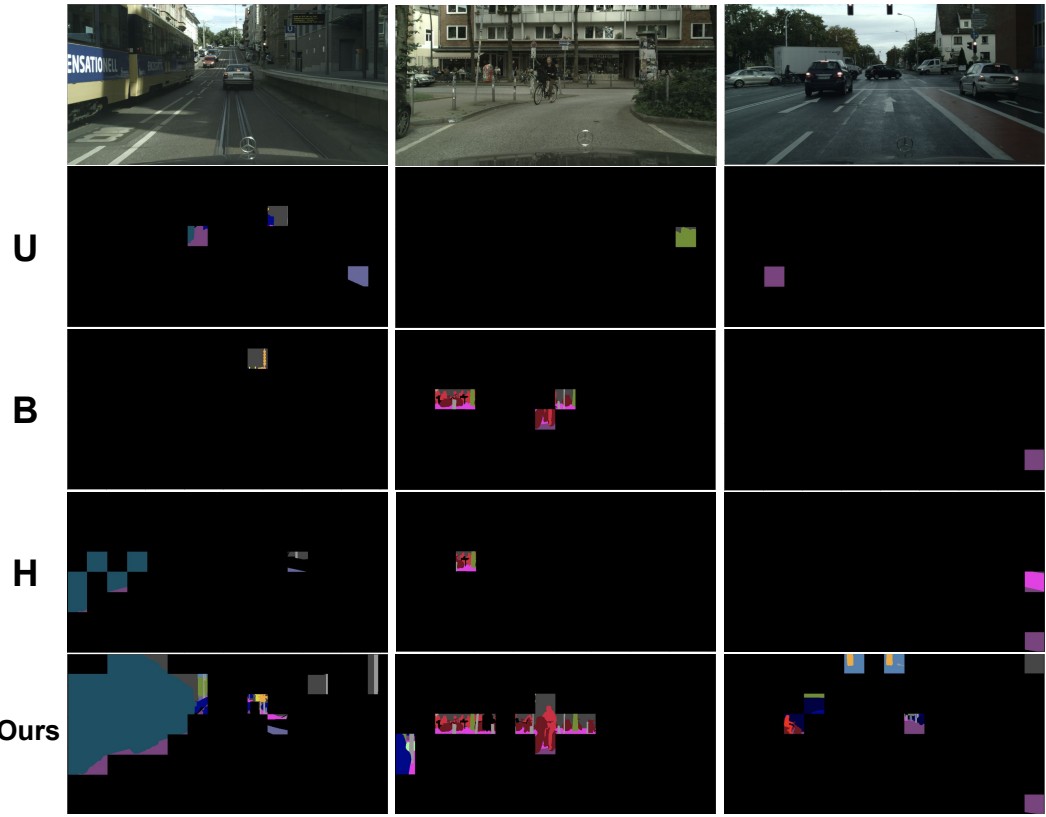

Figure D.1: Qualitative results in Cityscapes after running the active learning algorithm with a budget of 2k regions. The first row consists on input images, the second shows the what **U** picks, the third, **B**, the fourth **H**, and the last row shows what our method picks. Best viewed in color.

| State | Pool size | | |
|---|---|---|---|
| | **20** | **100** | **200** |
| **U** | $54.62 \pm 0.60$ | $54.92 \pm 0.59$ | $55.15 \pm 0.64$ |
| **H** (Shannon, 1948) | $57.41 \pm 0.17$ | $57.55 \pm 0.60$ | $57.48 \pm 0.96$ |
| **B** (Gal et al., 2017) | $56.73 \pm 0.20$ | $56.99 \pm 0.32$ | $56.44 \pm 0.77$ |
| **Ours - 1H** | $56.89 \pm 1.22$ | $57.29 \pm 0.67$ | $57.62 \pm 0.96$ |
| **Ours - 3H** | $57.65 \pm 0.74$ | $58.10 \pm 1.16$ | $57.65 \pm 1.30$ |
| **Ours - 3H+KL** | $57.67 \pm 0.92$ | $58.95 \pm 0.59$ | $\mathbf{59.18} \pm 0.62$ |

Table E.1: Contribution to the validation mean IoU performance [%] of Cityscapes dataset, for a budget of 4K and for each of the components of our state representation, compared to the baselines. Mean and standard deviation of 5 runs is reported.

## E.2    REGION VS. FULL IMAGE ANNOTATION

In this subsection, we analyze the effect of asking for labels in regions instead of full images and the effect of the number of regions per step. We compare the validation IoU when asking for pixel-wise labels for entire images versus pixel-wise labels for small regions. In the first case, we ask for one image at each step and, for the latter, we ask for 24 regions per step (pixel-wise, equivalent to one image). As it is shown in Table E.2, asking for entire image labels has similar performance for all methods, that resemble Uniform performance when asking for region labels. This indicates that, in order to select more informative samples, it is useful to split the images into patches (crops) and be able to disregard regions that only contain over-represented classes of the dataset.

## E.3 INFLUENCE OF STEP REGIONS

Empirically, our selector network is quite robust to the number of regions per step, as seen in Table E.3. Therefore, we select 24 regions for CamVid, the one that yielded best results. This is more efficient to train than taking one region per step.

| | U | H | B | Ours |
|---|---|---|---|---|
| Full im. | $69.64 \pm 0.33$ | $69.46 \pm 0.15$ | $69.66 \pm 0.21$ | $69.44 \pm 0.22$ |
| 24 R | $70.35 \pm 0.71$ | $70.40 \pm 0.65$ | $70.63 \pm 0.77$ | $\mathbf{71.85 \pm 0.68}$ |

Table E.2: Comparison between labeling a full image and 24 non-overlapping square regions (pixel-wise, equivalent to a full image), for different methods. Performance is measured in terms of validation mean IoU performance [%] in CamVid dataset, for a budget of 0.5k. In the first row, results for "full im.", one entire image is labeled at each step (region size equal to the size of the image). In the second row, "24 R" results for labeling 24 regions at each step. Pool size selected as the one that performed better, out of 10, 20, 50 and 100. Results are reported with the mean and standard deviation of 5 runs.

| Regions per step | Val IoU [%] |
|---|---|
| 1 | $71.10 \pm 0.75$ |
| 12 | $70.93 \pm 0.70$ |
| 24 | $\mathbf{71.85 \pm 0.68}$ |
| 36 | $71.24 \pm 0.49$ |
| 48 | $71.25 \pm 1.17$ |
| 72 | $71.20 \pm 0.53$ |

Table E.3: Results of varying the number of regions to be labeled at each step by our method. Performance is measured in terms of validation mean IoU performance [%] in CamVid dataset, for a budget of 0.5k Results are reported with the mean and standard deviation of 5 runs.

