# OpenReview forum: "Reinforced active learning for image segmentation"
_ICLR.cc/2020/Conference — Accept (Poster)_

### Official Review · AnonReviewer2 · 2019-10-24
**Official Blind Review #2**

**Rating:** 6

**Review:**

# Summary #
The paper works on active learning for semantic segmentation, aiming to annotate as few "blocks/patches" as possible while training a strong model. The authors proposed to learn a query policy via Q learning, and design states and actions specifically for segmentation. The experimental results show that the learned policy can attend to informative patches and rare classes to learn a model faster and efficiently.

# Strength #
S1. The paper is well-motivated; the references are quite sufficient.
S2. The paper clearly states the challenges when applying RL algorithms like Q-learning to image segmentation, which should serve as good guidance for other future work.

#Weakness/comments#
W1. The writing of the technical part can be strengthened.  The authors deferred the state and action design entirely to the supplementary, while they are the main contributions to the paper.

W2. The proposed algorithms seem to be highly time-consuming. The actions require pairwise comparison, and at every step, the models need to evaluate all the validation images to get the reward.

W3. If I understand correctly, the authors use part of the training data D_T, D_S (of an existing dataset) together with the validation data D_R to learn the policy, and then use the learned policy to select patches from the remaining training data D_V to train the segmentation model. I have two questions.
1) The labeled data involved in policy training is indeed quite large (validation plus part of the training, D_S + D_T + D_R). Does it mean that to learn a good active learning policy we indeed need a large number of labeled data?
2) Since (D_S, D_T, and D_R) are used to learn the policy, they should be treated as available training data for segmentation that all the compared algorithms can use without spending the budget. In other words, all the compared algorithms (U, H, B) should use those data to fine-tune a pre-train segmentation network before they start to acquire data from D_V. It would be great if the authors can clarify this.

W4. In applying the policy for selecting patches from D_V, do the authors update the model once on the selected patches, or do the authors train with them for multiple iterations together with other previous selected patches? Since deep neural nets are known to forget what has been learned (i.e., catastrophic forgetting), it's better if the author could clarify this.

W5. The authors include an upper bound in Fig. 4; however, I didn't find the explanation. Why the proposed methods can outperform the upper bound with all the training data, even with only 24% of data?

#Rebuttal#

Please discuss W1-W5.

- Annotating an entire image is definitely easier to annotators than annotating patches. Could the authors discuss how to design an active learning algorithm by selecting informative images to annotate, and maybe compare to such a method?

- Can the authors discuss Figure 3 more? As H is based on maximum entropy, why is it outperformed by the proposed method?

- There is no explicit mechanism to prevent that the k actions select similar patches. Can the authors provide more discussion?

# Post rebuttal
The authors responded to most of my concerns. I'd like the authors to incorporate all their responses into the manuscript or the appendix so that future readers can better understand the concepts and details.  I would like to raise the score to borderline (4 or 5). I modified my scores to weak accept (6) since there is no option in between.

One concern I still have is W3. 2). Given only 360 images are available, it might be inappropriate to use D_T + D_S (roughly 160 images?) to pre-train baseline models and then use D_R with "200" images for validation (early stopping). It will be more appropriate to use, for example, 70% of 360 images for training. This is supported by that many modern datasets use a much larger training set than the validation set. Therefore, I would highly suggest the authors redoing the baseline methods; otherwise, future work that re-splits the data (this is totally valid!) from the 360 images might easily achieve higher accuracy.

The authors must also reorganize the paper, taking W1 into account.

**Experience Assessment:**

I have read many papers in this area.

**Review Assessment: Checking Correctness Of Derivations And Theory:**

N/A

**Review Assessment: Checking Correctness Of Experiments:**

I carefully checked the experiments.

**Review Assessment: Thoroughness In Paper Reading:**

I read the paper at least twice and used my best judgement in assessing the paper.

---

> ### Author Response · Authors · 2019-11-08
> **Reply to reviewer 2**
>
> We thank the reviewer for the feedback and for raising those interesting discussion points, to be addressed in the following text.
>
> W1. We agree that the explanation about state and action design could be complemented with the information provided in the supplementary material. We will consider this point and modify the manuscript accordingly.
>
> W2. The model needs to obtain the reward (and evaluate the model in the reward set D_R) when the DQN is being trained. However, note that this is not required at the deployment step, when we actually test the algorithm in the D_V split.
> The algorithm has been designed to avoid human labor, that is costly and time consuming. Given that, computing action representations do need pairwise comparisons with other unlabeled and labeled regions -- and therefore it raises the computation time. To alleviate this computation cost, one could subsample the points used to do pairwise comparisons, or clustering them and taking just the centers.
>
> W3. 1) In Cityscapes, the subset of labeled data that we consider available is  composed of 360 images (12% of the entire training set). From those images, a local-validation set (D_R) is composed of 200 images. We use this split as if it were our validation split, while we consider the real Cityscapes-validation split as our test set. This way, we ensure that the real validation split is only used to report the final results, and not to find hyperparameters for any of the baselines or our model.
> The DQN algorithm, similarly to most deep learning and RL algorithms, requires some amount of training data to learn its task. Considering this, we think that the size of the labeled subset is not that big, compared to the entire collection of unlabeled data.
>
> W3. 2) D_T is indeed used to pre-train the segmentation network for all baselines and our model. We argue that the size of D_S (10 images) is small enough to not do a significant impact on any model. D_R is treated as a local-validation set  (which all methods have access to) to do early-stopping and in the case of our method, for the reward.
>
> W4. Once a new batch of labeled data is acquired, the segmentation network is trained only with this newly acquired batch, and for just one step. However, once the budget is met, the network is trained until convergence with all the labeled data.
> Considering preliminary experiments, we thought this could be the best way of training the segmentation network, for two reasons. First, if we were to train with all labeled data at each step, the segmentation network would see many more times the early labeled batches than the last ones, resulting in an imbalanced way of presenting the data to the model. Second, one would not be able to separate the improvement due to the newly labeled batch of regions, since it could be washed out by the entire collection of labeled data so far.
>
> W5. To clarify the caption of Figure 4, the dashed line corresponds to the 96% of the performance of the segmentation model, having access to all labeled data. Methods can go over it, because it does not represent the 100% performance of the fully supervised upper-bound. The dashed line is a reference to compare against the fully supervised model, and it is actually not an upper-bound. We will make the caption of Figure 4 more clear.

---

> > ### Author Response · Authors · 2019-11-08
> > **Continuation of reply to reviewer 2**
> >
> > Concerning other raised points:
> >
> > Labeling patches vs images - In order to label a patch, the user would have access to the entire image, but only asked to label a region. The user therefore would always have full context, but would be able to limit the (labor-intensive) pixel-level labelling to a fairly small region. As such the amount of work is much less than labelling the entire image. Moreover, Table E.2 provides a study on selecting entire images versus patches. As seen in the table, all methods benefit from using patch labeling instead of labeling entire images. In addition, our method does better than the baselines when selecting regions.
> >
> > Explaining Figure 3 - The entropy in this figure shows the class distribution of the final labeled set, for different methods. A class distribution of labeled samples can be computed after counting the class frequency of the chosen patches, after each AL method is done (Figure C.1).Therefore, an entropy measure can be computed over those class distributions, which is what Figure 3 shows. Higher entropy means a more uniform class distribution in the set of labeled patches, that directly translates to a less imbalanced labeled dataset. The “maximum entropy” baseline takes the patches with highest entropy of the predictions, but it has nothing to do with the entropy over the final class distribution of the selected regions.
> >
> > K actions selecting similar patches - Indeed, there is no explicit mechanism to avoid this, since these actions are parallel. We think of K as a trade-off parameter. If K was 1, there would be no actions taken in parallel (and no room to select similar patches). However, there is a price to pay in terms of computation and speed. The approach becomes faster and less computationally expensive as K increases. However, the possibility of getting similar patches increases -- which can be bad or beneficial (if more patches of a given class are needed, selecting more of that will also help the model).

---

### Official Review · AnonReviewer1 · 2019-11-04
**Official Blind Review #1**

**Rating:** 6

**Review:**

This work presents a method to use active learning to control the labeling process to gather training data in the context of a semantic segmentation application. In particular, authors define actions, states and rewards, and slightly modify DQN, to learn a policy to select informative image regions to obtain pixel labels. The proposed strategy contains several novelties related to the model and the application domain. Furthermore, the paper is clear and well written.

One weak part of the proposed method is the predefined split of each image into 24 regions that are used to define the population of sampling candidates. I believe that, for the case of semantic segmentation, the arbitrary definition of the boundary of each region might degrade performance. This because for this application the real object boundary is highly informative. Actually, this could be a major reason to explain that the proposed method only presents a modest boost in performance with respect to baselines, which is clear only for the under-represented categories.

A point that it is not clear to me is the initial training of the segmentation network. Also, it is not clear how is the process to train the segmentation network after the inclusion of each new batch of labeled data (do the segmentation network is trained incrementally with just the new data?). Finally, I believe it will be good to include information about the computational complexity of resulting model.

In summary, the idea of using AL and RL to control the labeling process in a semantic segmentation application is interesting and particularly relevant for this application. While the proposed method has some weaknesses, it is novel and it will be of interest to people interested in semantic segmentation. I rate the paper as weak accept.

**Experience Assessment:**

I have read many papers in this area.

**Review Assessment: Checking Correctness Of Derivations And Theory:**

I assessed the sensibility of the derivations and theory.

**Review Assessment: Checking Correctness Of Experiments:**

I assessed the sensibility of the experiments.

**Review Assessment: Thoroughness In Paper Reading:**

I made a quick assessment of this paper.

---

> ### Author Response · Authors · 2019-11-08
> **Reply to reviewer 1**
>
> We thank the reviewer for finding the work novel and of interest, and appreciate the interesting points that have been raised, that will be now addressed.
>
> We decided to use a patch-based strategy for two reasons: (i) to make experiments easier to reproduce and (ii) to reduce confounding factors that would be introduced if a mechanism for selecting irregular regions was involved in the experimental design. In practice, when the performance is poor for a given class, we observe that the algorithm selects adjacent patches, covering new examples of objects from the class of interest. Figure D.1 shows some examples of this behaviour, where a lot of adjacent patches of the bus are selected for the first image. The same happens in the second image; adjacent patches are selected to be labeled, where the person at the center with a bike appear, although they are split in several regions.
>
> The features of the regions (or patches) are extracted in the following way: first, the entire image is fed to the segmentation network; second, the final prediction is split in regions. Therefore, the information of the entire image has been captured by the segmentation network, and transferred to the individual regions. These individual regions are the ones used to compute state and action representations.
>
> However, we do agree that there could be better ways of defining the regions. In order to respect the actual boundaries of the objects, one could use a superpixel segmentation or object proposal algorithm. This could, however, be subject to the possible fake boundaries generated by the failures of this algorithm and it would also bring more complexity to the entire framework.
> We followed the work of [1] and used rectangular/squared regions to the entire framework slightly simpler. Finding which type of region splitting would work best is not in the scope of this work. We see it as an interesting future work.
>
> Improving the performance of a classifier on underrepresented classes is very important in applications like autonomous driving -- and our method was designed to address these settings. Indeed, the major improvements in performance happen to be in those underrepresented classes.
>
> The segmentation network has been pretrained equally for both our method and baselines. First, it has been trained with the GTAV dataset, to have a better calibration of the entropy features than having random predictions from an untrained network (for entropy, BALD and our method). Moreover, in order to ensure a fair comparison, the segmentation network (in every case) has been trained further with D_T (a subsplit of Cityscapes, used to train the DQN).
>
> Once a new batch of labeled data is acquired, the segmentation network is trained only with this newly acquired batch, and for just one step. Once the budget is met, the network is trained until convergence with all the labeled data.
> Considering early results, we thought this could be the best way of training the segmentation network, for two reasons. First, if we were to train with all data at each step, the segmentation network would see many more times the early labeled batches than the last ones. This could result in an imbalanced way of presenting the data to the model. Second, one would not be able to separate the improvement due to the newly labeled batch of regions, since it could be washed out by the entire collection of labeled data so far.
>
> [1] Radek Mackowiak, Philip Lenz, Omair Ghori, Ferran Diego, Oliver Lange, and Carsten Rother. CEREALS - cost-effective region-based active learning for semantic segmentation. BMVC, 2018.

---

### Public Comment · ~Samarth_Sinha1 · 2019-09-30
**Possibly interesting baseline**

Thanks for the interesting paper!

I agree with the chosen baselines, and comparing the proposed method to uncertainty based methods. But I am curious to know how well this paper would be able to compare to using a representation based method like Core-sets. There is some recent evidence that makes me believe that the notion of representation might be quite important for deep active learning, and I think this problem can also be formulated to be compared to Core-sets.

Another quick question I have is regarding the performance on BALD. In fig 4 it clearly outperforms uniform sampling on one dataset, and is significantly outperformed by random sampling on a different (but similar) dataset. Is there any insight as to why that might be the case? A few papers have been unable to get good performance using BALD in the deep learning setting [1,2], where it has significantly underperformed random sampling. I think touching upon that might be interesting sentence (or two) to the paper.

[1] Sener, Ozan, and Silvio Savarese. "Active learning for convolutional neural networks: A core-set approach." arXiv preprint arXiv:1708.00489 (2017).
[2] Sinha, Samarth, Sayna Ebrahimi, and Trevor Darrell. "Variational Adversarial Active Learning." arXiv preprint arXiv:1904.00370 (2019).

---

> ### Author Response · Authors · 2019-11-08
> **Under-performance of BALD method**
>
> Thank you for your comment and sorry for the late reply.
>
> Regarding the comparison with core-sets, we are having troubles with the commercial library Gurobi and we are currently looking into that. We apologize for not being able to provide a more specific answer for this part at the moment.
>
> What we observed is that BALD tends to oversample some classes. In Camvid, this might result in a combination of 1) low data regime 2) the few data that is labeled is highly skewed towards some classes. In Cityscapes, since there is more data available, this oversampling will hurt performance, but it might have enough samples for all classes to at least outperform the uniform baseline.
>
> As seen in Figure C.1 (in the Appendix),  in Cityscapes, BALD tends to oversample regions that contain some of the classes such as Building, Vegetation and Sky (which are not under-represented). Also, in Table C.1, we can see that the performance for these 3 classes is similar between BALD and uniform sampling. Then, we could say that this oversampling that BALD “wasted” part of the labeling budget for this task and it still does not benefit the performance. However, for some other classes, BALD has a higher frequency than uniform, improving mean IoU and placing itself in a better position than the uniform baseline.

---

### Author Response · Authors · 2019-11-15
**Summary of rebuttal so far**

We would like to summarize the points brought up by the two reviewers during the rebuttal so far.

The paper was found well written and clear (R1), well motivated and with sufficient references (R2). The work presented has been acknowledged to clearly present the challenges of RL algorithms in image segmentation (R2) and to present novelty in the method and application domain (R1). Moreover, the work is seen as a positive work for future research and of interest in the semantic segmentation domain (R1,R2).

We also addressed the concerns brought up by both reviewers, being the main points the following:  1) clarification for the data splits and training scheme used, 3) the choice of region type, which we clarified it is not in the scope of this work and provided further discussion.  A decision on how to split images was chosen based on previous work and to facilitate the exploration of the RL algorithm from this part forward.

Moreover, we remain open to further feedback and we thank reviewers again for their helpful reviews.

---

### Decision · Program_Chairs · 2019-12-19

**Decision:**

Accept (Poster)

**Comment:**

Authors propose a novel scheme to perform active learning on image segmentation. This structured task is highly time consuming for humans to perform and challenging to model theoretically as to potentially apply existing active learning methods. Reviewers have remaining concerns over computation and that the empirical evaluation is not overwhelming (e.g., more comparisons). Nevertheless, the paper appears to bring new ideas to the table for this important problem.